# Fetal Myocardial Expression of GLUT1: Roles of BPA Exposure and Cord Blood Exosomes in a Rat Model

**DOI:** 10.3390/cells11203195

**Published:** 2022-10-11

**Authors:** Leonardo Ermini, Maurizio Mandalà, Laura Cresti, Sofia Passaponti, Laura Patrussi, Luana Paulesu, Kent Thornburg, Francesca Ietta

**Affiliations:** 1Department of Life Sciences, University of Siena, 53100 Siena, Italy; 2Department of Biology, Ecology and Earth Sciences, University of Calabria, 87036 Rende, Italy; 3Center for Developmental Health, Knight Cardiovascular Institute, Oregon Health & Science University, Portland, OR 97239, USA

**Keywords:** BPA, exosomes, placenta, fetal heart, GLUT1

## Abstract

Dietary exposure to Bisphenol A (BPA), an industrial chemical present in food containers, affects nutrient metabolism in the myocardium of offspring during intrauterine life. Using a murine model, we observed that fetal hearts from mothers exposed to BPA (2.5 μg/kg/day) for 20 days before mating and for all of the gestation had decreased expression of glucose transporter-1 (GLUT1), the principal sugar transporter in the fetal heart, and increased expression of fatty acid cluster of differentiation 36 transporter (CD36), compared to control fetuses from vehicle-treated mothers. We confirmed the suppression of GLUT1 by exposing fetal heart organotypic cultures to BPA (1 nM) for 48 h but did not detect changes in CD36 compared to controls. During pregnancy, the placenta continuously releases extracellular vesicles such as exosomes into fetal circulation. These vesicles influence the growth and development of fetal organs. When fetal heart cultures were treated with cord blood-derived exosomes isolated from BPA-fed animals, GLUT1 expression was increased by approximately 40%. Based on our results, we speculate that exosomes from cord blood, in particular placenta-derived nanovesicles, could contribute to the stabilization of the fetal heart metabolism by ameliorating the harmful effects of BPA on GLUT1 expression.

## 1. Introduction

Maternal consumption of nutrient-rich food is required for normal fetal development; however, it is also the principal route of exposure to environmental contaminants. Bisphenol A (BPA) is an industrial chemical used in the production of polycarbonate plastics and epoxy resins contained in food containers [1]. The chemical is defined as an endocrine-disrupting chemical (EDC) due to its estrogenic activity and interference with endocrine pathways. Because of its widespread use in plastics, 90% of the population is exposed to detectable levels of BPA. There is a variable global distribution of BPA that ranges from 0.004 to 17 ng/m^3^ in some urban areas of several countries such as China, Japan, and the United States, to 0.001 and 0.03 ng/m^3^ in marine areas such as the Pacific and Atlantic Oceans [2]. In plasma samples taken from 150 people in Malaysia, the mean BPA level was 2.22 ± 9.91 ng/mL [3]. Levels were higher in females than in males.

In pregnant women, BPA is a potentially dangerous chemical for the growing fetus due to its ability to cross the placenta and reach the conceptus where it is found in amniotic fluid, placental tissue and cord blood [4,5]. The placenta is a transient organ through which all nutrients must pass to assure robust fetal development and growth. A large body of evidence associates BPA with morphological and functional changes in the placenta. BPA is also associated with epigenetically-derived detrimental changes in gene regulation and it contributes to the early onset of pregnancy and fetal disorders [6,7,8,9,10,11,12,13,14,15,16].

Recent epidemiological studies have shown associations between BPA levels in adult plasma with coronary and peripheral arterial disease [17,18,19]. Maternal BPA exposure alters the transcriptome and structural features of the fetal heart [20,21,22]. Moreover, exposure to this chemical during the prenatal period has been associated with altered metabolism that can lead to the development of cardiometabolic diseases [23,24]. In particular, BPA exposure during pregnancy in mice alters fetal glucose homeostasis and could be a risk factor for cardiovascular and diabetes development in the offspring [25].

Fetal heart metabolism is glucose-dependent, supported by a high expression level of the glucose transporter protein, GLUT1, and a low level of the fatty acid transporter protein CD36. After birth, the metabolism proceeds toward oxidative phosphorylation and fatty acids become the preferred fuel, whereupon the expression profile of the two nutrient transporters is reversed [26]. GLUT1 protein levels are mainly regulated by the transcriptional factors Sp1 and Sp3 during fetal heart development. Sp1 promotes the transcription of GLUT1 during fetal life and is strongly diminished in the early neonatal period. On the contrary, Sp3 has an inhibitory impact on GLUT1 expression during postnatal maturation of the myocardium; its repressive effect is stronger that the stimulatory effect of Sp1 [27].

Fetal heart development is finely regulated by several factors such as humoral signals and mechanical forces. The placenta controls many of those factors. Resistance in the placental vascular bed can alter fetal circulation and fetal heart development. Moreover, the placenta regulates substrate exchange between mother and fetus and the release of signals into both circulations [28]. Fetal glucose requirements depend on the ability of the placenta to transport glucose from the maternal blood via the activity of GLUT1, the dominant glucose transporter isoform in the placenta [29]. Several recent pieces of evidence have demonstrated that placentas release exosomes into both fetal and maternal circulations that orchestrate the maternal-fetal cross-talk often mediated by non-coding RNAs [30]. Interestingly, the alteration of the expression of some circulating exosomal miRNAs was identified in the blood of women pregnant with a fetus with congenital heart disease (CHD), and these miRNAs have been postulated to regulate fetal cardiac development [31].

Recent data demonstrated that miRNAs and other exosomal factors are important vehicles for metabolic organ cross-talk, in particular to maintain glucose homeostasis [32]. The exosomes can alter the levels of transporters involved in glucose uptake. For instance, cardiomyocytes can release exosomes that promote the levels of GLUT1 and GLUT4 to increase glucose uptake as well as exosomes released by hepatic stellate cells, therefore, showing that those exosomes are potential modulators of glucose metabolism and uptake in cells [33,34]. It was recently demonstrated that maternal exposure to a low dose of BPA (2.5 μg/kg/day) stimulates GLUT1 expression in rat placenta [10]. Additionally, exposure of human trophoblast and placental explants to BPA increases GLUT1 expression and glucose uptake in trophoblast cell lines [10,12]. Fetal exposure to high doses of BPA (5 ppm and 20 ppm) stimulates fibrosis in the heart and reduces the expression of cardiac troponin I [22]. However, the degree to which BPA affects fetal heart nutrient transporters at environmentally relevant concentrations is unknown [35]. To test the hypothesis that BPA regulates the expression of cardiac nutrient transporters, consistent with the evidence in the placenta, we investigated the impact of maternal dietary exposure to BPA in fetal rats.

## 2. Materials and Methods

### 2.1. Animal Model

All experiments were conducted following the European Guidelines for the care and use of laboratory animals (Directive 26/2014/EU) and were approved by the local ethical committee of the University of Calabria and the Italian Ministry of Health (n.74/2018-PR). Fetal hearts were isolated from the fetuses of pregnant rats euthanized on gestational day 20 as previously described [10]. Briefly, Sprague Dawley rats were separated into two groups and treated respectively with BPA at 2.5 μg/Kg/day BPA in ethanol (n = 6) or with the ethanol vehicle alone (Control, n = 6), as previously described [10]. BPA or vehicle was added to drinking water, based on daily water consumed and the body weight, for a month (virgin state) plus 20 days during pregnancy starting from the first day of gestation. Animals were then euthanized on the 20th day of pregnancy and fetal hearts (n = 14) (n = 7 male and n = 7 female) from control rats and n = 18 (n = 9 male and n = 9 female) from treated rats were collected and weighed. The heart specimens were also processed for immunohistochemistry and alternatively stored at −80 °C for subsequent protein extraction and western blot (WB) analysis as well as for triglycerides (TG) and glucose quantification. The fetal hearts from unexposed rats for in vitro experiments were directly cut with a vibratome and treated as described below.

### 2.2. Organotypic Cultures

Organotypic cultures of rat fetal hearts (n = 3 of separate experiments) were performed as previously described [36].

To investigate BPA’s direct action on the fetal heart, cultures were incubated for 48 h with 1 nM BPA (Sigma Chemical Co., Burlington, MA, USA), which corresponds to the concentration present in tissues and fluids during pregnancy [37] or with the vehicle ethanol (0.1%) in DMEM. The samples were then processed for immunofluorescence (IF) or WB analysis.

### 2.3. Exosome Isolation and Characterization

Exosome isolation and characterization were performed as described by Ermini et al. [38,39]. Briefly, cord plasma obtained from animals fed or not with BPA was centrifugated at 2500× *g*, at 4 °C for 30 min to remove apoptotic bodies. The supernatant was then further spun for 1 h at 15,000× *g* to eliminate the microvesicles and ultra-centrifugated at 200,000× *g* for 2 h. The obtained pellet of extracellular vesicles was suspended in PBS and filtered through a 0.22 µm filter. Exosome quality was then confirmed by TEM (Transmission electron microscopy), NanoSight particle analyzer (Malvern Instruments Ltd., Malvern, UK), and WB for CD63 (standard exosome marker) as well as for syncytin 1 (a marker of placental derived exosomes). To examine exosome uptake by heart specimens, the nanovesicles were labeled using the PKH67 Fluorescent Cell linker Kit (Sigma, PKH67GL-1KT) as previously described [39] as a general membrane stain and the marked vesicles were used as a probe at different times on the organotypic cultures of fetal rat heart. The uptake was then determined using a Zeiss LSM700 (Zeiss, Oberkochen, Germany) for confocal microscopy. Organotypic cultures of sliced fetal hearts were then incubated for 16 h with 2.0 × 10^6^ ExoBPA and ExoCt exosomes (isolated respectively from cord plasma of BPA-fed and control animals). The treated cultures were then processed for IF and WB analysis.

### 2.4. RNA Isolation and Real Time PCR

Fetal hearts were subjected to RNA isolation using a TRIzol reagent (Life Technologies, Invitrogen, Carlsbad, CA, USA, Cat. No. t9424), and cDNA was obtained using a high fidelity reverse transcription (RT) kit (Applied Biosystems, Foster City, CA, USA). Gene expression levels of ESR1 were established by Real Time PCR using commercial pre-developed primers (Bio-Rad, Hercules, CA, USA; Cat. No qRnoCID0009588) for Syber green evaluation. mRNA levels were normalized using GAPDH and 18S as housekeeping. The expression level was calculated by the 2^−^^ΔΔCt^ [40].

### 2.5. Glucose Quantification

The intracellular glucose content was measured using the Amplex Red Glucose Assay Kit^®^ (Thermo Fisher Scientific ^®^, Waltham, MA, USA).

### 2.6. TG Quantification

The triglyceride content of fetal hearts was measured using a Triglyceride Quantification Kit (Sigma) according to the manufacturer’s protocol.

### 2.7. Immunohistochemistry

Fetal rat heart samples were first fixed in 4% (*v*/*v*) buffered formalin (pH 7.2–7.4) and dehydrated through increasing alcohol steps before embedding in paraffin. The 4 μm slides obtained by microtome cutting were heated to 60 °C for 10 min and rehydrated through decreasing alcohol steps and finally washed in Tris-buffered saline (TBS) (20 mM Tris-HCl and 150 mM NaCl, pH 7.6). Subsequently, the slides were pre-incubated in 3% (*v*/*v*) H_2_O_2_ for 15 min to block endogenous peroxidase reaction, and then incubated with Protein Block Serum-Free (0.25% (*w*/*v*) casein in PBS solution, with stabilizing protein and 0.015 mol/L sodium azide) (Dako, Santa Clara, CA, USA) to block the non-specific binding of antibodies. After blocking, the slides were incubated overnight at 4 °C with primary antibodies (diluted in TBS). The slides were washed and pre-incubated with Peroxidase anti-Peroxidase (PAP) complex (Dako), an amplifier of the reaction, and then incubated for 1 h with the appropriate peroxidase-conjugated secondary antibody (HorseRadish Peroxidase, HRP) diluted in TBS. The reaction was detected by Diaminobenzidine (DAB) and the sections were counterstained with Mayer’s Hematoxylin, mounted on an aqueous medium (Merck, Whitehouse Station, NJ, USA) and examined under an optical microscope. In negative controls, the primary antibodies were substituted by the appropriate normal isotype antibody in TBS. Antibodies were tested for specificity by western blot.

### 2.8. Immunofluorescence

The immunofluorescence microscopy on the organotypic culture of rat fetal heart slices was performed according to Patrussi et al. [41]. Briefly, heart slices were moved to slides and incubated for 30 min at room temperature with PBS containing 4% paraformaldehyde. The samples were then washed and permeabilized for 30 min with PBS containing 0.01% Triton X-100. Slices were then stained with primary antibodies, washed, and incubated with secondary antibodies conjugated with fluorochrome. Nuclei were stained with 1 μg/mL DAPI (Sigma Chemical Co.) in PBS. The sections were then washed with PBS and mounted using a fluorescence mounting medium. Images were acquired on a Zeiss LSM700 confocal microscope.

### 2.9. Transmission Electron Microscopy (TEM)

TEM analysis of exosomes was performed as described in Ermini et al. [38].

### 2.10. Nanoparticle Size Measurements

The concentration and diameter of cord blood exosomes were measured using a NanoSight NS 300 particle analyzer (Malvern Instruments Ltd., Malvern, UK).

### 2.11. Western Blot

Western blot analysis of fetal rat heart tissues and explants were performed as described in Ermini et al. [12]

### 2.12. Antibodies Used in the Research

Primary antibodies used in the research include mouse monoclonal anti-CD36 (sc-7309, SantaCruz, Dallas, TX, USA; WB: 1:1000, IC: 1:100), mouse monoclonal anti-CD63 (MX-49.129.5, SantaCruz; WB: 1:500), mouse monoclonal anti-GAPDH (G8795, Sigma; WB: 1:2000), rabbit polyclonal anti-GLUT1 (PA5-16793, Thermo Fisher Scientific^®^; WB: 1:1000, IC: 1:100, IF: 1:100), mouse monoclonal anti-GLUT4 (2213; Cell Signaling, Danvers, MA, USA; WB 1:1000), rabbit polyclonal anti-LC3B (2775, Cell Signaling; WB: 1:1000), mouse monoclonal anti-GLUT3 (sc-74497, SantaCruz; WB: 1:500) rabbit polyclonal anti-PARP (9542, Cell Signaling; WB: 1:1000), mouse monoclonal anti-p21 (2946, Cell Signaling; WB: 1:2000), rabbit polyclonal anti- ERVW-1 (SAB2108833, Sigma; WB: 1:1000). HRP-conjugated secondary antibodies were obtained from Bio-Rad (Bio-Rad.) and used at a concentration of 1:3000. For IF, Alexa Fluor^®^ 488 donkey anti-rabbit IgG (A21206) were purchased from ThermoFisher Scientific^®^ (ThermoFisher Scientific^®^).

### 2.13. Statistical Analysis

Densitometric analyses were performed using Image J software. Data are expressed as mean ± SEM and were analyzed with GraphPad Prism 7.0 (GraphPad Software, Inc., San Diego, CA, USA). The obtained data were statistically examined using an Unpaired Student’s *T*-test. Significance was accepted at *p* < 0.05.

## 3. Results

### 3.1. Maternal Dietary Exposure to BPA Alters Nutrient Transporter Expression in Fetal Heart Cardiomyocytes

Pregnant rats were exposed to BPA (2.5 μg/Kg/day) in drinking water, and fetal hearts were collected as described in the methods section. Fetal hearts were then processed for immunohistochemistry and western blot (WB) analysis. As shown in Figure 1, GLUT1 protein levels decreased in fetal hearts exposed to BPA and CD36 protein levels increased. (Figure 1A–C). The immunohistochemical analysis revealed that, compared to the control, BPA treatment reduced GLUT1 mainly in the epicardial region of the heart (Figure 1A), while the increase in CD36 affected all heart tissue. In addition to GLUT1, the prevalent glucose transporter in the fetal heart, we performed protein expression analysis on GLUT4 (the primary transporter in the postnatal/adult heart) and GLUT3 known for its higher affinity for hexose compared to GLUT1 and GLUT4 [42]. As can be observed in Appendix A, the levels of GLUT4 were significantly decreased in the fetal heart of BPA-fed animals, and GLUT3 tended to be increased (Appendix A).

Interestingly, the glucose content in fetal hearts from BPA-fed animals was similar to levels in control hearts (Figure 2A), which is partially explained by the results obtained on GLUT3.

On the other hand, the increase of the fatty acid transporter resulted in an increased flux of fatty acids into the cardiomyocyte and a corresponding significant accumulation of triglycerides (TG) in the fetal heart (Figure 2B). Nevertheless, changes in nutrient transporters and lipid accumulation were not accompanied by increases in heart weight (Figure 2C).

Excessive lipid accumulation could cause the formation of lipotoxic compounds [43]. We observed that lipid levels did not change some pathways related to cell apoptosis, autophagy, or cell cycle regulation as revealed by the WB for Poly (ADP-ribose) polymerases (PARP), protein light chain 3 (LC3), and p21 (Figure 2D, Appendix A). However, further analysis needs to be performed to confirm the lack of alteration in fetal myocyte cell fate.

To confirm BPA activity in the fetal heart, we performed a qPCR analysis of the estrogen receptors alpha (ESR1), an exposure marker to the chemical [44]. As we can observe in the plot in Appendix A, the BPA-exposed hearts showed a significant upregulation of the expression of ESR1.

### 3.2. Direct BPA Exposure Decreases GLUT1 Expression in Fetal Heart Cardiomyocytes

To examine whether the changes in nutrient transporter levels in fetal hearts were due to a direct action of BPA, we performed in vitro studies on fetal heart organotypic cultures established from unexposed mothers. Treatment with 1 nM BPA reduced GLUT1 protein expression as occurred in vivo exposure (Figure 3A,B). WB revealed low levels of CD36, and no changes were observed, suggesting that other mediators are involved in vivo for some effects of BPA on the fetal heart (Appendix A).

### 3.3. Maternal BPA Exposure Alters the Cargo of Cord Blood Nanovesicles to Antagonize its Effect on the Fetal Heart

To investigate whether maternal BPA exposure could facilitate the vesicular cross-talk between mother and fetal heart, we isolated exosomes from cord plasma of BPA-fed and control rats. Nanoparticle Tracking (Figure 4A) and TEM (see Appendix A) analysis confirmed that we isolated small extracellular vesicles of the diameter of exosomes (around 150 nm). The exosomal nature of the isolated vesicles is further confirmed by WB analysis for CD63, TSG101, and CD9 (exosome markers) and H3 (apoptotic body marker) (data not shown). A nanoparticle tracking analysis revealed no significant changes in the concentration and diameter of the extracellular vesicles derived from control and BPA-exposed animals (Figure 4A,B). The characterization of cord exosomes from BPA-fed rats showed higher levels of Syncytin 1 protein (a well-known placental exosome marker) compared to control rats. (Figure 4C).

To study whether the cord exosomes were able to be up taken and act in recipient heart cells, nanovesicles were labeled with a green fluorescent dye and incubated with fetal heart organotypic cultures from unexposed animals. We observed that the nanovesicles were incorporated in fetal heart cells as detected by the measure of mean intensity fluorescence of the treated samples and colocalization staining using Rab7 (see Appendix A), a marker of late endosomes [45]. More importantly, fetal heart cultures treated with exosomes from BPA-fed rats (ExoBPA) showed significantly higher GLUT1 and CPT1 (the fatty acid mitochondrial transporter) levels compared to treatment with exosomes from control animals (ExoCt) (Figure 4D), suggesting that nanovesicles derived from BPA exposure contain signals that counteract the direct effect of the chemical.

## 4. Discussion

In the present study, we showed that maternal dietary exposure to an environmentally relevant concentration of BPA altered the nutrient transporter expression in fetal rat myocardium. In particular, in vitro experiments proved that BPA directly decreases GLUT1 levels in cardiomyocytes.

The GLUT family compromises fourteen known members divided into three classes and varies in tissue distribution, localization, substrate specificity, and kinetics [46,47]. Class I Facilitative Glucose Transporters, in particular GLUT1 and GLUT4, are the prevalent glucose transporters in the fetal murine heart, and their expression is strictly controlled during development [27]. GLUT3, another member of the class I Facilitative Glucose Transporters, has been found in both adult and fetal hearts [48]. Low levels of class III Facilitative Glucose Transporters mRNA such as GLUT8 are detected in heart muscle even if GLUT8 suppression didn’t impair mice growth or glucose homeostasis during embryonic development [49]. Fetal heart metabolism is different than in adults. Carbohydrates, including glucose, are the principal source of energy. Thus, glycolysis is an important source of ATP for fetal cardiomyocytes [50]. Nakano’s data [51] suggest that immature, proliferative cardiomyocytes take advantage of high levels of glucose to fuel nucleotide biosynthesis through the pentose phosphate pathway. They also argue that glucose deprivation induces cardiac maturation at genetic, morphological, metabolic, electrophysiological and biomechanical levels. This conclusion is based on innovative studies conducted on human embryonic stem cell-derived cardiomyocytes [51]. The degree to which these findings apply to normally maturing cardiomyocytes in vivo has yet to be determined. Our data suggested that maternal BPA exposure does not modify the fetal heart metabolic profile and phenotype even if glucose transporter levels change.

The placental transport of glucose, amino acids and lipids affect the supply of these circulating substrates in the fetus. Thus, plasma levels are the result of a balance between supply and extraction at the organ level. Our previous data showed that, unlike the suppression of GLUT1 in the fetal rat heart, maternal exposure to BPA increases GLUT1 expression in the placenta as well as the ratio between fetal and placental weights. These data suggest an enhanced efficiency for placental nutrient/glucose transport in fetuses exposed to BPA resulting in an increased fetal weight [10]. Our data indicate that fetal heart weight did not change following exposure to BPA. We can therefore assume considering our previous finding that exposure to the chemical alters the heart to body weight ratio. Recent evidence shows that elevating glucose levels increases CD36 expression and lipid deposition in renal tubular cells [52]. Furthermore, in a recent study in H9c2 cells, an immortalized cell line established from the ventricular part of a 13-day-old rat embryo [53], high glucose exposure induced the CD36 mRNA and protein expression as well as lipid accumulation within the cells in a time-dependent manner [54]. We can therefore speculate that the upregulation of CD36 in fetal hearts exposed to BPA in vivo was the consequence of increased maternal-fetal transport of glucose.

One might wonder why the glucose content of the myocardium was maintained when GLUT1 levels were suppressed. One possible explanation is the increase in other transporters. GLUT3 has a much higher affinity for hexose compared to GLUT1 and GLUT4 [27] Thus, the trend toward increased levels of GLUT3, in BPA-exposed hearts as well as the increased circulating concentrations of glucose, could explain the relatively normal glucose levels in the fetal heart despite reduced levels of glucose transporters. BPA is an endocrine disruptor chemical that can bind estrogen receptors [55]. The chemical interacts with high affinity with members of a subfamily of nuclear receptors named estrogen-related receptor gamma, which is highly expressed in the fetal heart and is a key regulator of cardiac myocyte maturation by sustaining the transition to the oxidative phosphorylation in the adult heart and negatively regulating anaerobic glycolysis [56,57,58]. Moreover, neonatal exposure in rats to Bisphenol A alters the expression of transcription factors Sp1 and Sp3 [59] which, as previously described, regulate the expression of GLUT1 during heart development and are pivotal in the estrogen receptor pathway. Thus, we speculate that BPA effects are associated with this nuclear receptor via Sp1 and 3 transcriptional factors. Further studies will address this assumption.

During pregnancy, the placenta continuously releases several factors into the maternal and fetal circulation, including RNA proteins, DNA and hormones. Moreover, syncytiotrophoblast cells secrete extracellular vesicles (apoptotic bodies, syncytial nuclear aggregates, microvesicles and exosomes [60]. Exosomes are small vesicles (50–150 nm that originate from the endosomal compartment, secrete into the extracellular environment and carry signals to recipient cells [61]. Exosomes play a central role in pregnancy, modulating several processes including the maternal immunological response and metabolic adaptations. In particular, exosomes have been repeatedly indicated as components of fetal-maternal communication during implantation and placentation, while modulating maternal reaction, maintaining cellular metabolic homeostasis, promoting fetal vasculogenesis, maternal uterine vascular adaptation and preparing the uterus for labor [62]. Several studies have demonstrated fetal-maternal trafficking of exosomes during pregnancy. In particular, placental exosomes could reach tissues on the fetal side of the placenta, influencing the growth and development of fetal organs [63].

Current findings suggest that BPA changes metabolic activities and the placenta secretion of various molecules [5,9,10,15,64]. We found that BPA did not alter the concentration and diameter of cord blood exosomes. However, we observed that exposure to BPA caused dysregulation of the protein cargo of placental exosomes (present in the pool of the cord exosomes) indicated by the significant increase of placental Syncytin 1. Our findings agreed with previously published data by other groups [65]. Interestingly, recent data showed that syncytin proteins in placenta exosomes are critical for interaction with target cells [66,67] suggesting that the BPA-triggered upregulation of syncytin 1 in exosomes could favor the vesicular communication between the placenta and cells in fetal organs.

In addition to proteins, placental exosomes transfer genetic information to target cells by regulating their metabolism and cell fate. Placental microRNAs (miRNAs) are selectively packaged in the exosomes, secreted into the maternal and fetal circulation and transferred to target cells where they regulate gene expression [68]. Recent evidence suggests that miRNAs regulate GLUT1 expression as well as cardiac mitochondrial function [69,70,71]. Moreover, BPA alters the expression of several small RNAs in the mouse placentas [68,72,73]. We can therefore speculate that, in a mother exposed to BPA, the placenta releases exosomes enriched with Syncytin 1 in addition to non-coding RNAs that may regulate fetal heart metabolism. This assumption is also corroborated by our findings regarding the increased expression of Carnitine palmitoyltransferase 1 (CPT1) in the fetal heart organotypic cultures treated with ExoBPA. CPT1 is a mitochondrial enzyme present in the outer membrane and is involved in the transport of long chain fatty acids into the mitochondria [74]. Its upregulation could contribute to the removal of fatty acid accumulation triggered by the increase of CD36 in the fetal hearts obtained by BPA-fed animals, therefore preserving the homeostasis of the fetal heart. Future efforts will focus on the characterization of exosomal miRNAs that can alter cardiac function and metabolism. However, we must point out that the effect that we observed in the in vitro experiments refers to acute exposure to the chemical. As we have seen, chronic exposure to BPA during pregnancy leads to a significant reduction in GLUT1. Therefore, we assume that in chronic exposure additional BPA-triggered factors are involved in the regulation of GLUT1 expression.

In summary (Figure 5), BPA exposure in prenatal life alters nutrient transporters in the rat fetal heart, causing an increase in fatty acid absorption. If these effects persist into postnatal life, chronic BPA exposure to a high-fat diet could cause cardiac lipid overload and lead to susceptibility to heart failure in adulthood. Furthermore, the data on cord blood exosomes suggest that signals, present probably in exosomes of placental origin, could partially ameliorate the harmful effects of BPA on the rat fetal heart. Thus, we hypothesize that the downregulation of GLUT1 would be more severe were it not for the modulating effects of exosomes.

The main aim of our in vitro experiment was to highlight the direct effects of BPA on the two nutrient transporters in heart tissue and in parallel to investigate the potential role of cord blood exosomes on the expression of these transporters. In future experiments, to confirm the exosome capacity to ameliorate the dangerous effects of BPA, we will further investigate, using in vivo and in vitro models, the uptake of placental exosomes from the fetal cardiomyocytes and their potential role as rescue factors.

## Figures and Tables

**Figure 1 cells-11-03195-f001:**
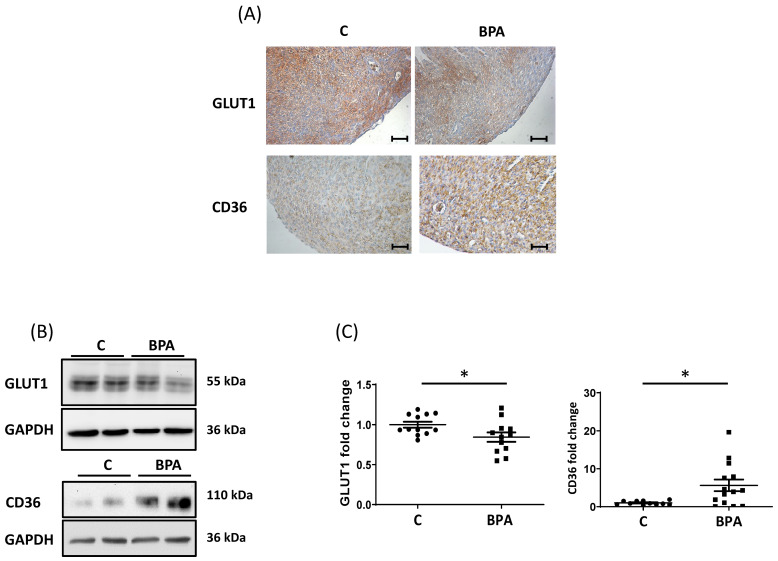
Tissue expression and localization of GLUT1 and CD36 in fetal rat hearts from maternal dietary exposure to BPA. (**A**) Representative immunolocalization of GLUT1 and CD36 in fetal hearts obtained from BPA feed and control rats. Magnification 20×. Scale bar = 50 μm (**B**) Representative WB for GLUT1 and CD36 in lysates of fetal hearts obtained from BPA feed and control rats. Glyceraldehyde-3-Phosphate Dehydrogenase (GAPDH) was used as a loading control. (**C**) Densitometry for GLUT1 [control (C, n = 12) vs. treatment (BPA, n = 12)] and CD36 [control (C, n = 11) vs. treatment (BPA, n = 14)] in lysates of fetal hearts obtained from BPA fed and control rats. GAPDH densitometry was used to normalize GLUT1 and CD36 values. Data are expressed as mean ± Standard Error of the Mean (SEM). The obtained data were statistically examined using an Unpaired Student’s *T*-test. Significance (*) was accepted at *p* < 0.05.

**Figure 2 cells-11-03195-f002:**
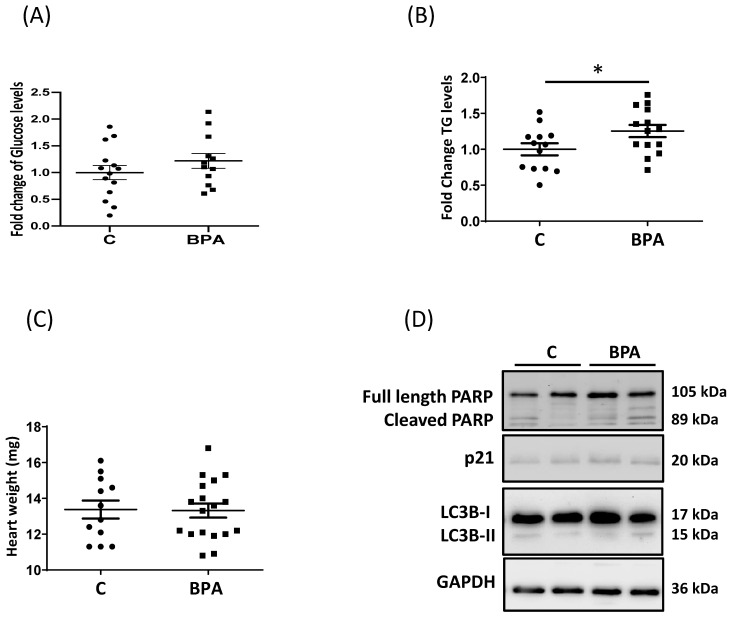
Weight, triglyceride, glucose levels and markers of pathways related to cell fate in fetal rat hearts from maternal dietary exposure to BPA. (**A**) Glucose levels in fetal hearts obtained from BPA-fed (n = 12) and control (n = 14) rats. Fetal hearts obtained from maternal dietary exposure to BPA or control were lysed and the glucose was quantified using Amplex Red Glucose Assay. (**B**) Triglyceride (TG) levels of the fetal heart obtained from BPA-fed (n = 14) and control (n = 13) rats. Fetal hearts obtained from maternal dietary exposure to BPA or control were lysed and the lipid was quantified using a Triglyceride Quantification Kit. (**C**) Fetal heart weight from BPA fed (n = 14) and control (n = 13) rats. Animals exposed or not to BPA were euthanized on the 20th day of pregnancy and fetal hearts were collected and weighed. (**D**) Representative WB for Poly (ADP-ribose) polymerases (PARP), protein light chain 3 (LC3), and p21 in lysates of fetal hearts obtained from BPA feed and control rats. GAPDH was used as a loading control. Data are expressed as mean ± Standard Error of the Mean (SEM). The data were analyzed using an Unpaired Student’s *T*-test. Significance (*) was accepted at *p* < 0.05.

**Figure 3 cells-11-03195-f003:**
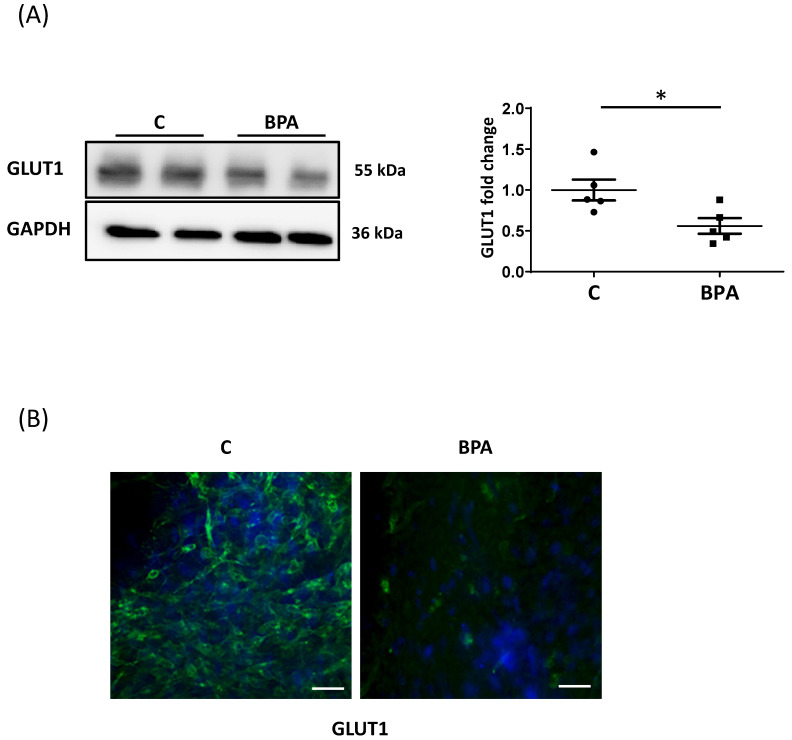
Effect of BPA on nutrient transporters in organotypic fetal heart cultures. (**A**) Representative WB for glucose transporter, GLUT1 and relative densitometry for GLUT1 in lysates of organotypic fetal heart cultures treated with 1 nM BPA or vehicle (ethanol, control) for 48 h (n = 3 of separate experiments). GAPDH was used as a loading control and to normalize GLUT1 densitometry. (**B**) Immunofluorescence of GLUT1 in cultured fetal cardiac tissue treated with 1 nM BPA or vehicle for 48 h. Scale bar = 50 μm Magnification: 40× (n = 3 of separate experiments). Data are expressed as mean ± Standard Error of the Mean (SEM). The obtained data were statistically examined using an Unpaired Student’s *T*-test. Significance (*) was accepted at *p* < 0.05.

**Figure 4 cells-11-03195-f004:**
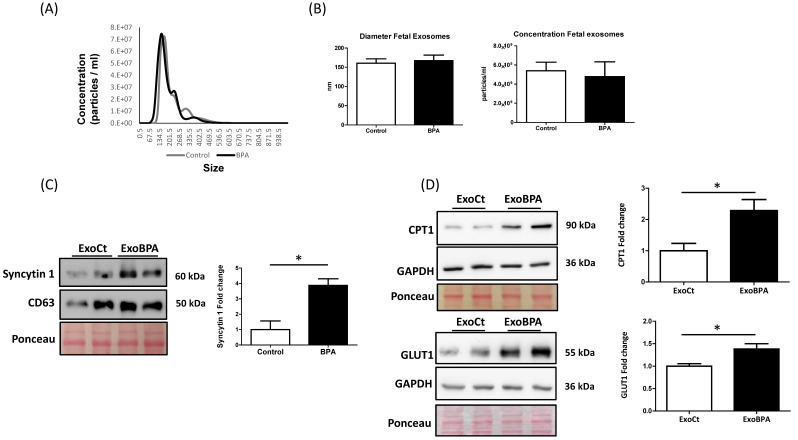
Effect of cord blood exosomes on nutrient transporters in organotypic fetal heart cultures. (**A**) Nanoparticle Tracking profile of cord blood exosomes obtained from BPA and control rats. Cord blood exosomes were isolated by ultracentrifugation and analyzed with the NS300 equipped with Nanoparticle Tracking Analysis (NTA) software. (**B**) Diameter and concentration of exosomes obtained by Nanoparticle Tracking Analysis from BPA (n = 3) and control rats (n = 3). (**C**) Representative WB for Syncytin 1 and associated densitometry of cord exosomes obtained from BPA (n = 5) and control rats (n = 3). CD63 was used as a loading control and to normalize Syncytin 1 densitometry. (**D**) Representative WB for GLUT1 and fatty acid mitochondrial transporter (CPT1) on organotypic fetal heart cultures treated for 16 h with 2.0 × 10^6^ cord exosomes obtained from BPA (ExoBPA) and vehicle (ExoCt) feed rats (n = 3 of separate experiments). GAPDH was used as a loading control and to normalize GLUT1 densitometry. Data are expressed as mean ± Standard Error of the Mean (SEM). The obtained data were analyzed using an Unpaired Student’s *T*-test. Significance (*) was accepted at *p* < 0.05.

**Figure 5 cells-11-03195-f005:**
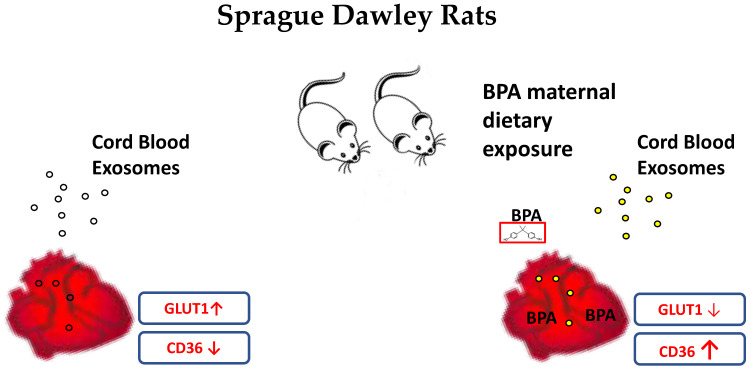
Schematic representation of the effects of maternal dietary exposure to BPA on rat fetal hearts. Maternal dietary exposure to BPA directly and indirectly alters the levels of nutrient transporters on fetal cardiomyocytes. Moreover, the chemical triggers the release of placental exosomes that contain signals that try to antagonize its harmful effects on fetal cardiomyocytes.

## Data Availability

The raw data supporting the conclusions of this article will be made available by the authors, without undue reservation.

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
