# Peer review of "Fetal Myocardial Expression of GLUT1: Roles of BPA Exposure and Cord Blood Exosomes in a Rat Model"

_cells, 2022, doi:10.3390/cells11203195_

Round 1
Reviewer 1 Report
This study found that gestational BPA exposure suppresses GLUT 1 and increases CD36 expression in fetal rat heart tissues. These effects were partially rescued by cord blood-derived exosome exposure in vitro cultured heart tissues. This study is relatively novel in exploring BPA's effects on fetal heart development. However, a major revision is needed before considering publishing in the cells.
The main concerns are listed below
Title.
- Since this study only uses rats as a model, the title should make it specific.
Abstract
- Some background was missing, such as BPA, exposure period, GLUT1, and exosomes.
Introduction
- The information on the environmentally relevant concentration of BPA is missing;
- The relationship between GLUT1 and fetal heart development/maturation (associated mechanisms, such as molecules and signaling pathways) needs to be introduced.
- The potential links between exosomes, glucose transporters (not only GLUT1), and heart development/maturation are needed.
- The link between placental function and fetal heart metabolisms and function needs to be introduced.
M&M
- The concentration of BPA in the drinking water is missing.
- How did you translate the BPA dose from in vivo study to the in vitro study?
- There are no total fetal numbers and fetal sex information.
- The full name of TEM?
Results
- It is confusing that there are only two samples per treatment group in all WB figures, and some of the WB bands were not quantified (Fig 2 D)
- There are no bands for both control and BPA group samples. How did you compare the differential expression of CD36 (Fig 3 A )?
- There is no Nuclear staining to show the overall cell distribution, and the scale bar was also missed.
- Figure 4 C does not have internal control. And the Syncytin 1 band is also a concern since at least one control group sample has no band, which doesn't make sense.
- Why does internal control of Figure 4 D use ponceau instead of GAPDH?
- The rat model was used in this study, but not humans. Please correct figure 5.
Discussion
- The results were not fully discussed in the discussion section.

Author Response
Title.
- Since this study only uses rats as a model, the title should make it specific.
We thank the reviewer for pointing this out. We have modified the title of the manuscript
Abstract
- Some background was missing, such as BPA, exposure period, GLUT1, and exosomes.
We agree and have updated the abstract with the lacking background.
Introduction
- The information on the environmentally relevant concentration of BPA is missing;
- The relationship between GLUT1 and fetal heart development/maturation (associated mechanisms, such as molecules and signaling pathways) needs to be introduced.
- The potential links between exosomes, glucose transporters (not only GLUT1), and heart development/maturation are needed.
- The link between placental function and fetal heart metabolisms and function needs to be introduced.
We thank the reviewer for the insightful comments. We have modified the introduction following the suggestions.
M&M
- The concentration of BPA in the drinking water is missing.
The BPA or ethanol was added in drinking water, on the basis of daily water drink and the body weight to arrive at the final concentration for the BPA of 2.5 μg/Kg/day.
- How did you translate the BPA dose from in vivo study to the in vitro study?
The effective dose of 1 nM BPA corresponds to the concentration present in tissues and fluids during pregnancy (10.1093/humrep/17.11.2839)
- There are no total fetal numbers and fetal sex information.
We have inserted the lacking information in the paragraph Animal model.
- The full name of TEM?
We have inserted the full name of TEM (Transmission electron microscopy) at the first appearance of the abbreviation.
Results
- It is confusing that there are only two samples per treatment group in all WB figures, and some of the WB bands were not quantified (Fig 2 D)
We apologize if our blots in the manuscript did not show all the samples that we have quantified. The blots are just representative of the profile of the semi quantification. We added the quantification of the blots of figure 2 D in supplementary figure 2.
- There are no bands for both control and BPA group samples. How did you compare the differential expression of CD36 (Fig 3 A )?
This observation is correct. The CD36 is barely expressed in fetal heart and we just qualitative notice no changes in the faint band. We moved the blots in the supplementary figure 3.
- There is no Nuclear staining to show the overall cell distribution, and the scale bar was also missed.
This was an oversight. We have added the missing nuclear staining and scale Bar in the figure 3.
- Figure 4 C does not have internal control. And the Syncytin 1 band is also a concern since at least one control group sample has no band, which doesn't make sense.
Thank you for your nice reminder. We have stripped the blot and incubated again the membrane with a new batch of antibody and we were able to observe also the lacking band. The new blot is inserted in figure 4 C. CD63 is a widely recognized and popular exosomal marker used also for ELISA exosome quantification and therefore as an internal control of exosomes. Furthermore, the exosomes are loaded in equivalent concentrations thanks to the quantification with Nanosight. We have added also the ponceau image of the blot as another internal control.
- Why does internal control of Figure 4 D use ponceau instead of GAPDH?
We thank the reviewer for pointing this out. We have inserted the GAPDH blots as a loading control in Figure 4 D.
- The rat model was used in this study, but not humans. Please correct figure 5.
We thank the reviewer for the correct comment. We have modified figure 5 as suggested.
Discussion
- The results were not fully discussed in the discussion section.
We thank the reviewer for pointing this out. We have inserted a paragraph regarding the GLUT family in rat fetal heart, comments on the GLUT3 and CPT1 data, and inserted new comments regarding the possible pathway activated by BPA.

Reviewer 2 Report
The manuscript of Ermini et al. builds on and expands the data on BPA impact on rat pregnancy previously published by the same authors: the data are interesting and intriguing.
Major issues
1. It is not clear whether the authors have analyzed the entire fetal heart or just the (left or right?) atria or the (left or right) ventricles. Please clarify! If possible, provide a comparative analysis of (left/right) atria and ventricles!
2. mRNA quantification was normalized to GAPDH and 18S; please provide data showing the appropriateness of the usage of these two genes as endogenous markers in these particular experiments! Please provide the sequences of the primers used in the RT qPCR experiments.
3. GLUT 1 and GLUT4 are not the only transporters expressed in the murine fetal heart. Please provide data on the expression of other GLUT isoforms (at least GLUT8!). This would be important for explaining the unexpected lack of change in glucose levels in the BPA-treated hearts.
4. PARP, LC3, and p21 WB quantifications are definitely not enough to claim (the lack of) changes in apoptosis, autophagy, or cell cycle... . Please tone down your statement or bring forth further experimental arguments.
5. Please explain the choice of ESR1 as a BPA exposure marker of the fetal hearts (out of a wide array of genes shown to be regulated by BPA) vs. the measurement of unconjugated BPA.
6. The exposure of cultured hearts to cord-derived exosomes from BPA-exposed animals leads to upregulation of GLUT1; assuming the uptake of exosomes (marginally demonstrated by data shown in Figure S3) also occurs in vivo, this phenomenon should lead to normalization of GLUT1 level. In this respect, two sets of data would be relevant for the biological significance: a) if possible, a quantification of the exosome uptake phenomena? b) a rescue experiment of GLUT1 expression in BPA-treated explants with cord-derived exosomes from BPA-exposed animals.
Minor issues
1. Please check for typos.
2. I do not understand the role of TG data in the overall flow of the manuscript.
3. The discussion should also include a paragraph on GLUT isoforms expression in the murine fetal heart.
Author Response
Point-by-point responses.
Reviewer 2
- It is not clear whether the authors have analyzed the entire fetal heart or just the (left or right?) atria or the (left or right) ventricles. Please clarify! If possible, provide a comparative analysis of (left/right) atria and ventricles!
We thank the reviewer for pointing this out. GLUT1 is expressed in cells of the myocardial tissue while endothelial cells of the access of pulmonary veins were negative. We observed that compared to the control, BPA treatment decreases the GLUT1 in the epicardium, the most external layer of the ventricular wall, while the effect on CD36 was more distributed across the tissue We have inserted this statement in the result part.
- mRNA quantification was normalized to GAPDH and 18S; please provide data showing the appropriateness of the usage of these two genes as endogenous markers in these particular experiments! Please provide the sequences of the primers used in the RT qPCR experiments.
Recent data have indicated that during heart development or maturation, the expression levels for some housekeeping genes are significantly altered (10.1186/gb-2002-3-7-research0034). However, we have not compared the expression of ESR1 during heart development. We have investigated the expression of ESR1 in fetal heart samples at the same development phase but treated with BPA or with the vehicle as a control. Therefore, GAPDH and 18s are suitable for housekeeping. We have used commercial pre-developed primers (Bio-Rad, Cat. No qRnoCID0009588).
- GLUT 1 and GLUT4 are not the only transporters expressed in the murine fetal heart. Please provide data on the expression of other GLUT isoforms (at least GLUT8!). This would be important for explaining the unexpected lack of change in glucose levels in the BPA-treated hearts.
We thank the reviewer for the insightful comment. We have investigated the expression of GLUT3, another class 1 glucose transporter that is present in fetal and adult heart and has a much higher affinity for hexose compared to GLUT1 and GLUT4. We observed a positive not significant increase of GLUT3 in the fetal heart obtained by BPA-fed animals compared to the control one partially that could explain the relatively normal glucose levels in the fetal heart despite reduced levels of the main glucose transporters. We have inserted the data in figure 1 supplementary.
- PARP, LC3, and p21 WB quantifications are definitely not enough to claim (the lack of) changes in apoptosis, autophagy, or cell cycle... . Please tone down your statement or bring forth further experimental arguments.
As suggested, we have toned down the statement and declared that further analysis needs to be performed to confirm the lack of alteration in fetal myocyte cell fate.
- Please explain the choice of ESR1 as a BPA exposure marker of the fetal hearts (out of a wide array of genes shown to be regulated by BPA) vs. the measurement of unconjugated BPA.
It is difficult to quantify the active form of BPA because the chemical is quickly transformed into BPA-glucuronide which is the inactive form more water-soluble, and easily removed by excretion via urine. However, we have confirmed the presence and the activity of BPA on the heart of the treated animals by markers of exposure such as the upregulation of the estrogen receptors.
- The exposure of cultured hearts to cord-derived exosomes from BPA-exposed animals leads to upregulation of GLUT1; assuming the uptake of exosomes (marginally demonstrated by data shown in Figure S3) also occurs in vivo, this phenomenon should lead to normalization of GLUT1 level. In this respect, two sets of data would be relevant for the biological significance: a) if possible, a quantification of the exosome uptake phenomena? b) a rescue experiment of GLUT1 expression in BPA-treated explants with cord-derived exosomes from BPA-exposed animals.
We appreciate the reviewer’s suggestions.
a) We have quantified the exosome uptake by measuring the mean intensity fluorescence of the fetal organotypic culture after 3 hrs and 6 hrs of incubation with the labeled exosome. We have inserted this new data in the supplementary figure 5.
b) We agree with the reviewer that rescue experiments of GLUT1 expression would be helpful. However, the main aim of our in vitro experiment was to highlight the direct effects of BPA on the two nutrient transporters in heart tissue and in parallel to investigate the potential role of cord blood exosomes. The result that emerges is that BPA directly regulates only the expression of GLUT 1. We have therefore modified the title of the manuscript and inserted a final paragraph about the future studies.
Minor issues
- Please check for typos.
We have checked the manuscript for editing errors.
- I do not understand the role of TG data in the overall flow of the manuscript.
CD36 level increase could result in a raised flow of fatty acids into the cardiomyocyte, leading to excessive lipid accumulation (in the form of triacylglycerol; TG). Excessive lipid accumulation can result in the formation of lipotoxic compounds such as long-chain acyl-CoA (LCACoA), ceramides, and diacylglycerides, which are correlated with myocardial contractile dysfunction and may promote insulin resistance. (Biochimica et Biophysica Acta (BBA) - Molecular and Cell Biology of LipidsVolume 1861, Issue 10, October 2016, Pages 1450-1460).
The discussion should also include a paragraph on GLUT isoforms expression in the murine fetal heart.
We have inserted a paragraph regarding the GLUT family in murine fetal heart in the discussion.

Round 2
Reviewer 1 Report
The authors addressed my comments.
Reviewer 2 Report
Although I do not entirely agree with the authors on the GAPDH and 18S normalization issue, I think the manuscript can be accepted in its present form.